# Albumin-Albumin/Lactosylated Core-Shell Nanoparticles: Therapy to Treat Hepatocellular Carcinoma for Controlled Delivery of Doxorubicin

**DOI:** 10.3390/molecules25225432

**Published:** 2020-11-20

**Authors:** Nayelli Guadalupe Teran-Saavedra, Jose Andrei Sarabia-Sainz, Enrique Fernando Velázquez-Contreras, Gabriela Ramos-Clamont Montfort, Martín Pedroza-Montero, Luz Vazquez-Moreno

**Affiliations:** 1Departamento de Investigacion en Polimeros y Materiales, Universidad de Sonora, Bulevar Luis Encinas y Rosales s/n, Colonia Centro, Hernosillo, Sonora 83000, Mexico; naye_krebs@hotmail.com (N.G.T.-S.); enrique.velazquez@unison.mx (E.F.V.-C.); 2Departamento de Investigacion en Física, Universidad de Sonora, P.O. Box 5-088, Hermosillo, Sonora 83190, Mexico; martin.pedroza@unison.mx; 3Centro de Investigacion en Alimentacion y Desarrollo, A.C. Carretera Gustavo E. Aztiazaran 46, Hermosillo, Sonora 83304, Mexico; gramos@ciad.mx (G.R.-C.M.); lvazquez@ciad.mx (L.V.-M.)

**Keywords:** doxorubicin, drug delivery systems, hepatocellular carcinoma, asialoglycoprotein receptor, lactosylated albumin, core-shell nanoparticles

## Abstract

Doxorubicin (Dox) is the most widely used chemotherapeutic agent and is considered a highly powerful and broad-spectrum for cancer treatment. However, its application is compromised by the cumulative side effect of dose-dependent cardiotoxicity. Because of this, targeted drug delivery systems (DDS) are currently being explored in an attempt to reduce Dox systemic side-effects. In this study, DDS targeting hepatocellular carcinoma (HCC) has been designed, specifically to the asialoglycoprotein receptor (ASGPR). Dox-loaded albumin-albumin/lactosylated (core-shell) nanoparticles (tBSA/BSALac NPs) with low (LC) and high (HC) crosslink using glutaraldehyde were synthesized. Nanoparticles presented spherical shapes with a size distribution of 257 ± 14 nm and 254 ± 14 nm, as well as an estimated surface charge of −28.0 ± 0.1 mV and −26.0 ± 0.2 mV, respectively. The encapsulation efficiency of Dox for the two types of nanoparticles was higher than 80%. The in vitro drug release results showed a sustained and controlled release profile. Additionally, the nanoparticles were revealed to be biocompatible with red blood cells (RBCs) and human liver cancer cells (HepG2 cells). In cytotoxicity assays, Dox-loaded nanoparticles decrease cell viability more efficiently than free Dox. Specific biorecognition assays confirmed the interaction between nanoparticles and HepG2 cells, especially with ASGPRs. Both types of nanoparticles may be possible DDS specifically targeting HCC, thus reducing side effects, mainly cardiotoxicity. Therefore, improving the quality of life from patients during chemotherapy.

## 1. Introduction

Doxorubicin (Dox) is an anthracycline considered one of the most potent and broad-spectrum chemotherapeutic agents [1,2] in treating a wide range of many childhood and adult malignancies [3,4]. Dox has been used in oncologic practice since its discovery more than 50 years ago [3,5] and is still one of the most common and effective antineoplastic drugs currently in use [4,6]. The anticancer effects of Dox are divided into three different mechanisms. The intercalation of Dox with double-stranded DNA, the stabilization of the topoisomerase II α (TOP2A)–DNA cleavage complex by forming DNA adducts and the induction of oxidative stress [7,8]. Despite the advent of monoclonal antibody and targeted tyrosine kinase-based therapies, anthracyclines are still about 50% used against various neoplasms [4,9]. Doxorubicin is routinely prescribed in combination with other agents to treat various types of cancer. These include breast, lung, gastric, pancreatic, ovarian, liver and hematologic malignancies [10,11]. Dox’s acceptable use has been impeded by toxicities such as hematopoietic suppression, extravasation, nausea, vomiting and alopecia. However, the most feared side-effect is cardiotoxicity, limiting its use [8,12]. There are two types of incidence of Dox cardiotoxicity, acute and chronic. The incidence of acute cardiotoxicity is approximately 11% and comes about during within 2–3 days of its administration [13,14,15]. On the contrary, the incidence of chronic cardiotoxicity is much lower, with an estimated of about 1.7% and it is usually evident within 30 days of administration of the last dose [13,16]. 

These cardiac pathologies can range from asymptomatic ventricular dysfunction, to tachycardia and arrhythmias, cardiomyopathy, myocardial infarction, and congestive heart failure, being referred to as the most notorious and well-studied cardiovascular toxicities [3,13]. It currently, it is not possible to predict which patients will be affected by these side effects or adequately protect patients who are at risk for suffering from them [4]. The available treatment for cardiomyopathy has shown no improvement in patients [13]. Fortunately, better pharmacotherapies and interventions have been generated for the prevention of cardiotoxicity. Nonetheless, much work is required to validate the clinical usefulness of these new approaches proposed [8].

Because of this, targeted drug delivery systems (DDS) are currently being explored as alternative methods of conventional treatments in an attempt to effectively direct Dox to the specific lesion site and reduce its systemic side effects [17].

Targeted DDS can transport the drug to the site of action; thus, avoid contact of the drug with vital tissues and minimize side effects [18,19]. It has been reported that DDS protects the drug from rapid degradation or elimination, increasing the drug concentration in the tissue of interest [18]. This causes the drug dose to be lower compared to the administration of a free drug. 

Recent studies show that nanoparticles have a great potential as drug carriers. Inside of the variety of types, biopolymer-based nanoparticles are mostly used to develop targeted DDS [20]. This is due to integrating with a biological system without eliciting an immune or toxic response [18,20]. Therefore, they are an ideal material for biomedical applications. Additionally, it has been reported that the specific targeting of nanoparticles towards the site of interest may be accomplished in two ways, passive or active [18,21]. 

Passive targeting is mainly possible by the enhanced vascular permeability and retention (EPR). This can lead to macromolecules accumulation, increasing the concentration in the tumor 70-fold [21]. In contrast, active targeting employs strong interaction such as ligand-receptor or other molecular recognition to confer more specificity to targeted DDS. This last targeting is the one that shows the greatest potential, because if nanoparticles are designed for active targeting, passive targeting occurs first followed by active targeting. Targeted therapy applying nanoparticles, specifically to kill cancer cells has been reported. For example, Shao et al., 2007 [22] described that carbon nanotubes functionalized with antibodies and in combination with phototherapy properties of the complex, can lead to a new class of molecular delivery and cancer therapeutic systems. Furthermore, the affinity between ligand and receptor can be explored not only to reach solid tumors, but also to capture circulating malignant cells, such as in metastasis process [23].

Previously, we reported on the synthesis and characterization of lactosylated (Lac) albumin (BSA) nanoparticles (BSALac NPs), in addition to their specific recognition by the asialoglycoprotein receptor (ASGPR), which is practically exclusively and the most abundant receptor in liver cells [24]. This information is very important because hepatocellular carcinoma (HCC) is the fourth most common cause of cancer-related death worldwide with an incidence rate similar to that of the mortality rate [25]. Thus, these results insight into an increasingly popular nano-vehicle that has been potentially explored as a DDS targeted to HCC. 

The present work aimed to synthesize and characterize albumin (tBSA)/lactosylated albumin (BSALac) core-shell nanoparticles with two concentrations of crosslinker (glutaraldehyde), low (LC) and high (HC), that have the controlled release capacity of Dox (LC and HC tBSA/BSALac-Dox NPs). Furthermore, to evaluate the specific biorecognition by ASGPRs present on the HCC cell line (HepG2 cells), which is a cell line with more ASGPRs currently reported [26].

## 2. Results and Discussion

### 2.1. Characterization of the Nanoparticles

Previously, LC and HC tBSA/BSALac-Dox NPs were prepared using a desolvation system as previously described. Different concentrations of crosslinker were included to evaluate their impact on encapsulation efficiency [24]. Control nanoparticles that were synthesized without a BSALac shell were used to compare them with nanoparticles with BSALac shell. Thus, four different nanoparticles were obtained: LC tBSA/BSALac-Dox NPs, HC tBSA/BSALac-Dox NPs and as controls, LC tBSA-Dox NPs and HC tBSA-Dox NPs.

#### 2.1.1. Size, Zeta Potential and Encapsulation Efficiency of the Nanoparticles

The particle size, polydispersity index (PDI) and zeta potential of NPs are summarized in Table 1. The size of LC tBSA/BSALac-Dox NPs and HC tBSA/BSALac-Dox NPs was 257 ± 14 nm and 254 ± 14 nm, respectively. In turn, LC tBSA-Dox control NPs and HC tBSA-Dox NPs had smaller diameters than lactosylated nanoparticles. This could be evidence of the BSALac shell may increase the particle size. These sizes are large enough to resist rapid renal clearance and accumulate selectively at the tumor site via enhanced permeability and retention (EPR) effect [27]. This can contribute to a higher level of tumor cell cytotoxicity in an in vivo system. All NPs presented PDI values <0.2, considered to have a homogeneous narrow distribution [28]. Control NPs presented zeta potentials of −32.0 ± 0.5 mV and −31.0 ± 0.5 mV, while lactosylated NPs showed −28.0 ± 0.1 mV and −26.0 ± 0.2 for LC and HC, respectively. The main influence of the negative charge of the albumin NPS is the large number of amino acids with carboxyl residues compared to the amine groups of amino acids [29]. In the glycation reaction of BSA with lactose, only the amino groups participate, likewise glutaraldehyde as a cross-linker reacts with the amino groups to stabilize the NPs formed. In this sense, an intensification of the negative charge in lactosylated NPs would be expected, which is not evident when observing the Z potential results (−32.0 ± 0.5 mV and −31.0 ± 0.5 mV for non lactosylated NPs, and −28.0 ± 0.1 mV and −26.0 ± 0.2 for lactosylated NPs). However, the slight change in Z potential levels of lactosylated NPs may be due to the lactose is considered a neutral carbohydrate [30]. Therefore, lactosylation of BSA it would not have a high influence on the Z potential of NPs. On the other hand, it is important to highlight that negative values indicate that clearance by the reticuloendothelial system (RES) could be reduced due to the low absorption of plasma proteins on the surface of the nanoparticles [28].

The encapsulation efficiency (E.E.) of Dox was calculated as the percent ratio of the actual amounts of Dox incorporated in the nanoparticles and the total initial amounts of Dox (Table 1). The % E.E. of Dox from LC tBSA-Dox NPs, LC tBSA/BSALac-Dox NPs were 71.8 ± 1.3, 73.4 ± 0.8, respectively, while both HC tBSA-Dox NPs and HC tBSA/BSALac-Dox showed higher % E.E. (89 ± 2 and 91 ± 2, respectively). High values of EE% for dox loaded-albumin nanoparticles were previously reported by Motevalli et al., 2019 [31] where EE% were greater than 84%, and Thao et al., 2017 [32] with EE% for doxorubicin above to 81%. In this work, all types of nanoparticles showed a high drug encapsulation efficiency, probably due to the interaction of Dox with the BSA core. The hydrophobicity/hydrophilicity of Dox is pH-dependent. In fact, Dox at pH 7 or higher has a hydrophobic character. Additionally, reports show that albumin is a natural carrier of hydrophobic molecules [33,34]. Thus, hydrophobic interactions can increase the drug encapsulation properties.

#### 2.1.2. Stability of the Nanoparticles

The nanoparticles’ stability was evaluated by the change in particle size and zeta potential in a period of eight days (Figure 1). Throughout storage, the nanoparticle systems did not show flocculation or coalescence, and remained stable. Moreover, the particle size and the zeta potential slightly increased after storage. A similar tendency was reported by Choi and Meghani et al., [35]. with BSA, transferrin-modified BSA (Tf-BSA) and hyaluronic acid-modified BSA (HA-BSA) nanoparticles. In addition, functionalized nanoparticles were more stable than non-functionalized. Qi et al., 2010 [33] reported that nanoparticles with dextran/chitosan shell and BSA/chitosan core-Dox nanoparticles were stable for 30 days. These data coincide with LC tBSA/BSALac-Dox NPs and HC tBSA/BSALac-Dox NPs (Figure 1B,D respectively) were slightly more stable than LC tBSA-Dox NPs and HC tBSA-Dox NPs (Figure 1A,C respectively).

#### 2.1.3. Morphological Characterization of the Nanoparticles

The nanoparticles’ morphology was observed by scanning electron microscopy (SEM). The nanoparticles showed a homogeneous size distribution and shaped spherical with a smooth surface (Figure 2). The average diameters obtained by SEM for all nanoparticles synthesizer were around to 220–240 nm.

Nanoparticles with BSALac shells showed a slightly larger size than nanoparticles without shells. This information was also observed by DLS (Table 1). However, SEM and DLS data presented slightly different values for particle size. DLS provides particles’ data swollen in solution, while SEM shows the particles dried on the surface [33].

### 2.2. In Vitro Release Studies

In vitro release studies were evaluated within 96 h using a dialysis membrane immersed in PBS at pH 7.2 (Figure 3). Free Dox solution was used as a control. As it was expected, the LC tBSA-Dox NPs, LC tBSA/BSALac-Dox NPs, HC tBSA-Dox NPs and HC tBSA/BSALac-Dox NPs exhibit slower releases than the diffusion of free Dox. During the first 24 h, the diffusion of free Dox was approximately 80% while the nanoparticles systems were close to 20%. The accumulative release percent of HC tBSA-Dox and HC tBSA/BSALac-Dox NPs were slower than LC tBSA-Dox NPs and LC tBSA/BSALac-Dox NPs. This biphasic release pattern characterised by an initial release (burst effect) followed by a slower release was reported for drug-loaded albumin nanoparticles [36]. Thus, nanoparticles are potential systems for drug release control, with the capacity to improve HCC therapy compared to the free drug [37].

Also, the amount of crosslinker influences the release rate of Dox (Figure 3). A higher degree of crosslinking of the nanoparticles causes a slower release of DOX. However, the release profile of all nanoparticles obtained could not be adjusted to any mathematical model. Other systems, using sodium ferulate loaded BSA NPs (SF-BSA NPs) reported a similar behavior Li et al., 2008 [38].

Furthermore, nanoparticles with BSALac shell showed a slight decrease in the release percent compared to the nanoparticles without BSALac shell. Synthesis of nanoparticles with a BSA core crosslinked with more glutaraldehyde and BSALac shell formed nanoparticles with sustained Dox release phase for several days. This is important, because the slow release of chemotherapy drugs makes them efficient toward cancerous cells [39,40] and improving patient acceptance due to the reduction of the amount and application time.

### 2.3. Hemocompatibility of the Nanoparticles

The use of nanoparticles in biomedical applications, particularly in the pharmaceutical area, is crucial to test for toxicity be evaluated. Nanoparticles are intentionally engineered to interact with the cells of the body; thus, it is essential to ensure that the nanocarriers are not causing any adverse effect. Blood cell contact with foreign agents, such as nanoparticles, can cause a various of hemolytic conditions in the body [41,42].

Blood compatibility (hemocompatibility) is an essential criterion to verify nanostructured systems’ safety and their null toxicity in the blood [42]. In this work, the nanoparticles’ hemocompatibility was determined by hemolysis assays and red blood cells (RBCs) viability assay.

#### Hemolysis Assay and RBCs Viability Assay

RBCs hemolysis and viability assay were evaluated at a concentration of 500 μg/mL of nanoparticles for 24 and 72 h. The percent of nanoparticles hemolytic activity was determined by evaluating the supernatant absorbance at 545 nm (hemoglobin) using UV-vis spectroscopy [41]. The LC tBSA-Dox NPs, LC tBSA/BSALac-Dox NPs, HC tBSA-Dox NPs and HC tBSA/BSALac-Dox NPs were non-hemolytic after 24 and 72 h of incubation with RBCs (Figure 4). The percent of hemolysis from all nanoparticles was below 2%, indicating the hemocompatibility of nanoparticles. Nanostructured systems are considered non-toxic if the hemolysis value is below 5%, according to the ASTME2524-08 standard [43].

Complementary studies to determine the hemotoxic activity of all nanoparticle samples, the RBCs viability assay were tested [44]. Trypan blue was used to estimate the number of viable RBCs by counting intact cells that have not been stained with trypan dye under the microscope [45] (Figure 4). All nanoparticles’ results showed almost 100% cell viability (<2% loss of membrane continuity). These results corroborate the absence of a considerable hemolytic effect (>20%), during 24 and 72 h.

Consequently, the negligible hemolytic activity and high RBC viability suggest that de nanoparticles obtained in this work are hemocompatible, and this gives a step forward to continue with further studies for biological use, specifically in drug delivery. 

### 2.4. Specific Biorecognition Assays

The targeting of nanoparticles by a ligand with the ability to be recognized by a specific receptor, generates a high efficiency in the delivery of drugs to specific tissue [46]. Therefore, when synthesizing ligand-functionalized nanoparticles, it is crucial to assess specific biorecognition. LC tBSA/BSALac-Dox NPs and HC tBSA/BSALac-Dox NPs have galactose residues in their shell, potentially recognizable by the lectin-type receptors (carbohydrate recognizer) [24]. In this work, the biorecognition assays were evaluated through the specific recognition of galactose residues by Ricinus communis Agglutinin I (RCA I) and specific cell uptake by HepG2 cells.

#### Evaluation of the Specific Recognition of Galactose Residues by RCA I

RCA I interaction with the galactose residues from nanoparticles was analyzed by enzyme-linked lectin recognition assay (ELLA) [24,47]. In Figure 5, the absorbances obtained from LC tBSA/BSALac-Dox NPs and HC tBSA/BSALac-Dox NPs were 0.20 ± 0.02 and 0.21 ± 0.03, respectively, demonstrating the biorecognition of lectin for galactose residues in its structure. In contrast, LC tBSA-Dox NPs and HC tBSA-Dox NPs showed basal absorbance of 0.05 ± 0.01.

Previously, we reported BSALac nanoparticles synthesized by alcoholic desolvation with the characteristic of being recognized with RCA I [24]. Diaz-Galvez et al., 2019, obtained similar results when using lactosylated graphene oxide (OGALac) was recognized by RCA I, corroborating that the synthesis allowed an appropriated functionalization [47]. These results indicate that galactose closed-ring structure (essential for RCA I interaction) was not modified by the synthesis.

### 2.5. In Vitro Cytotoxic Activity of the Nanoparticles

In vitro cytotoxicity of Dox loaded in either LC tBSA-Dox NPs, LC tBSA/BSALac-Dox NPs, HC tBSA-Dox NPs or HC tBSA/BSALac-Dox NPs and free Dox in HepG2 cells was assessed using a colorimetric assay (Figure 6) [47]. Nanoparticles without Dox (LC tBSA NPs, LC tBSA/BSALac NPs, HC tBSA NPs and HC tBSA/BSALac NPs) were evaluated as a control to determine the non-cytotoxicity of the nanocarrier. The cell viability obtained with control nanoparticles at different concentrations was similar to the cells viability that were not in contact with nanoparticles (Figure 6A). This shows that the synthesis of nanoparticles based on tBSA and BSALac is viable, generating biocompatible nanocarriers.

In Figure 6B,C the results showed that dox loaded in nanoparticles with BSALac shell (LC tBSA/BSALac NPs and HC tBSA/BSALac NPs) inhibited the viability and proliferation of HepG2 cells at low concentrations. The slightly more cytotoxicity of nanoparticles with BSALac shell compared with nanoparticles without BSALac shell, are the evidence of the targeted ability of lactose to HepG2 cells improved cell uptake, therefore, the cytotoxic effect [28]. This effect was previously reported by Quan et al., 2015 [48] where the cellular uptake study showed the internalization process for lactosylated NPs was energy-consuming and predominated by clathrin-mediated pathway and suggested that this nanoparticle transported the drug into hepatoma cells more effectively than non-lactosylated NPs. Lactosylated NPS enhanced cytotoxicity activity, which was correlated to pathway of cellular uptake. Further studies showing the different intracellular pathways with active and passive targeting are necessary.

In Table 2, show for Dox loaded in LC tBSA NPs, LC tBSA/BSALac NPs, HC tBSA NPs and HC tBSA/BSALac NPs the IC50 values obtaining 1.05 ± 0.13, 0.7 ± 0.09, 1.04 ± 0.15 and 0.59 ± 0.07, respectively, which were not significant. However, the IC50 was higher for free Dox (1.90 ± 0.42), showing statistical difference (*p* < 0.05) with all samples of Dox loaded in nanoparticles. This lower cytotoxic activity of free Dox may be due to self-defense mechanisms cells. Cancer cells show resistance to chemotherapeutic drugs, so the drug molecules eventually decrease their anticancer potential. On the other hand, the larger cytotoxic effect of Dox-loaded nanoparticle may be due to nano-carriers ability to evade cell defense mechanisms against to the drug, increasing their efficacy [49]. Thao et al., 2010 [32] reported BSALac nanoparticles loaded with Dox and paclitaxel (Pac) (Dox/Pac Lac-BSA NPs) obtained more effective cytotoxicity than non-lactosylated nanoparticles (Dox/Pac BSA NPs). This information suggests that DDS targeted to HCC is an alternative to evade drug resistance and, therefore, increase therapy efficiency.

#### Evaluation of Specific Recognition by HepG2 Cells

HepG2 cell (ASGP-R positive) and human cervical carcinoma (HeLa) (ASGP-R negative) cell cultures were exposed to synthesized nanoparticles and evaluated by confocal fluorescence microscopy after 30 min. Biorecognition, competence, and inhibition assays were performed to evaluate the specificity of nanoparticle-cell interactions (Figure 7).

HepG2 cells incubated with LC tBSA/BSALac-Dox NPs and HC tBSA/BSALac-Dox NPs presented detectable red fluorescence (Dox emitted) in the majority of the cells, indicative of galactose biorecognition by the ASGPR present. In contrast, null signal was detected in HeLa cells. Additionally, is important to point out that LC tBSA-Dox NPs and HC tBSA-Dox NPs showed reduced biorecognition by HepG2 cells, this could be attributed to the presence of gp60 receptors present in tumor cells, with the ability to recognize BSA [24].

To confirm specific carbohydrate biorecognition of LC tBSA/BSALac-Dox NPs and HC tBSA/BSALac-Dox competition and inhibition assays were performed using free lactose. In competition assays, a reduced fluorescence intensity was observed, signifying a specific interaction of the galactose residues with the receptor.

In the inhibition assays, the preincubation of HepG2 cells with free lactose shows a null biorecognition of the carbohydrate present in nanoparticles by the ASGPRs. Therefore, exposure of HepG2 cells to free lactose blocked ASGPR biorecognition and confirms that interaction was mediated by carbohydrates present in LC tBSA /BSALac-Dox NPs and HC tBSA/BSALac-Dox.

## 3. Materials and Methods 

### 3.1. Materials

All of the reagents used were analytical grade. Bovine serum albumin (BSA; 66.5 kDa and ~96%), D-lactose monohydrate (Lac), glutaraldehyde (25%), doxorubicin (Dox) hydrochloride, Dulbecco’s modified Eagle’s medium (DMEM), fetal bovine serum (FBS), penicillin/streptomycin (P/S), and [3-5-dimethylthiazol-2-yl]-2,5-diphenyltetrazolium bromide (MTT) were purchased from Sigma-Aldrich (St. Louis, MO, USA). *Ricinus communis* agglutinin I (RCA I) was obtained from the Vector Lab (Burlingame, CA, USA). Human liver cancer cells (HepG2 cells) and human cervical carcinoma cells (HeLa cells) were obtained from ATCC (Manassas, VA, USA). Unless specified, all other reagents and chemicals were purchased from Sigma-Aldrich (St. Louis, MO, USA). All the experiments were carried out using Type 2 pure water (0.18 μS cm^−1^).

### 3.2. Synthesis of Nanoparticles

The nanoparticles were prepared using a previously described method with minor modifications [24]. Before the colloid’s synthesis, the heat-treated and lactosylated BSA (tBSA and BSALac) were synthesized, as previously reported by Teran-Saavedra, et al., 2019 [24]. The procedure for obtaining nanoparticles consisted of generating a desolvation system to form four types of nanoparticles as shown in Table 3.

Figure 8 shows the schematic illustration of the procedure to obtain dox-loaded nanoparticles. Briefly, tBSA (10 mg) and Dox (500 μg) were dissolved in 1 mL deionized water (D.W.). Ethanol (3 mL) was added slowly dropwise into the mixture under constant stirring. Glutaraldehyde 8% (5 or 10 µL) was added to each corresponding sample (Table 3) to induce the crosslinking of tBSA molecules and stabilize the nanoparticles formed. The solutions were mixed at a low rotating speed at 25 °C for 5 h. The resulting nanoparticles were washed three times with deionized water and recovered by centrifugation (1644× *g* for 10 min at 15 °C). Low and high crosslinking tBSA nanoparticles with Dox (tBSA-Dox NPs) were obtained. Subsequently, the nanoparticles were coated with BSALac getting core-shell type nanoparticles (tBSA/BSALac-Dox NPs). For this, the nanoparticles (10 mg) were stirred in a solution of BSALac (2 mg/mL). After 5 min of stirring, glutaraldehyde 1% (2 µL) was added, and it was at low rotating speed at 25 °C for 5 h. Finally, the nanoparticles were rewashed under the conditions as mentioned earlier.

### 3.3. Characterization of Nanoparticles

#### 3.3.1. Size, Zeta potential, PDI and Stability of the Nanoparticles

The particle size (mean diameter), PDI and zeta potential of the obtained nanoparticles were measured by a Zetasizer Nano ZS90 (Malvern Instruments, Malvern, UK) operated with DLS. Data were collected at a scattering angle of 90° with the temperature maintained at 25 °C. Samples were diluted in PBS (1 mg/mL) at pH 7.2. The procedure to determine the stability consisted of generating nanoparticles (pH 7.2) were stored in PBS in a refrigerator for eight days. On days 0, 2, 4, 6 and 8, 1 mg/mL of nanoparticles was taken, and size and zeta potential were evaluated. Measurements were recorded as the average of three test runs.

#### 3.3.2. Nanoparticle Morphology

Nanoparticles were characterized by SEM (JEOL JSM-7800F, Akishima, Tokyo, Japan). SEM images were obtained using an acceleration voltage of 1.00 kV and images were obtained with a magnification of ×15,000. 

### 3.4. Encapsulation Efficiency and In Vitro Release Studies

To determine Dox in the nanoparticle encapsulation efficiency, the nanoparticles (10 mg) were dissolved in a 1 mL, 1:1 PBS at pH 7.2 and molecular trypsin solution. Then thoroughly shaken for 5 min and the nanoparticles were incubated at 37 °C for 24 h. The solution was centrifuged at 1644× *g* for 10 min. Dox in the supernatant was quantified directly by spectrophotometry at 480 nm [32]. The EE% calculation was determined with Equation (1).
(1)$$EE\%=\left[\frac{amount\;of\;encapsulated\;drug}{amount\;of\;total\;drug\;loaded}\right]\times 100$$

The in vitro release studies were determined using PBS at pH 7.2 as the release medium, at 37 °C. Dox loaded nanoparticles solution (2 mL) was dialyzed (MwCO 14 kDa) against release PBS (20 mL) with continuous agitation. At predetermined times, a 2 mL released medium was collected and the same medium was returned to dialysis. The absorbance at 480 nm was measured to analyze the Dox concentration in the release buffer. Each experiment was performed in triplicate. The accumulative release was expressed as percentage vs. initial loading amount at each time point [32,50].

### 3.5. Hematocompatibily

Voluntary healthy male donors from 20 to 30 years old were chosen for the hemocompatibility tests. The experiments were approved by the Ethics Committee of Sonora University, and complied with the principles expressed in the Helsinki’s declaration. All the participants signed an informed consent approved greed to the use of their blood in a hemolysis study and Red Blood Cells (RBCs) Viability Assay.

#### 3.5.1. Hemolysis Assay

Blood was collected into tubes with anticoagulant (BD Vacutainer EDTA). Blood was diluted in PBS to a 15:1000 μL relation, respectively. Then, the nanoparticles (500 μg/mL) were added to the solution of blood-PBS (1 mL). The assay was incubated for 24 and 72 h at 37 °C. Eventually, centrifuged at 2000× *g* for 1 min and the supernatants were collected to analyze of the extent of hemolysis by measuring hemoglobin’s absorbance of at 545 nm, using a spectrophotometer (Thermo Scientific Multiskan GO, Vantaa, Finland) [47]. The positive control was a sample treated with deionized water that causes complete hemolysis and negative control was an unlysed sample treated with PBS. The percentage of hemolysis was calculated with Equation (2).
(2)$$\%\;Hemolysis=\left[\frac{Sample\;treated\;with\;nanoparticles\;-\;Sample\;treated\;with\;PBS}{Sample\;treated\;with\;D.W.\;-\;Sample\;treated\;with\;PBS}\right]\times 100$$

#### 3.5.2. Red Blood Cells (RBCs) Viability Assay

The RBCs’ viability in the presence of nanoparticles was evaluated by the trypan blue dye exclusion test [45]. Failure to exclude trypan blue reflects a loss of plasma membrane integrity associated with cell lysis. The RBC was exposed to nanoparticles (500 μg/mL) for 24 and 72 h at 37 °C. After that, RBC suspension (10 μL) was mixed with trypan blue 0.4% (10 μL) and the cells were counted by the exclusion method. Cell viability was measured in a Neubauer chamber while using a microscope with 40× magnification. Finally, it was calculated by the Formula (3) and to eliminate variability, three replicates per concentration were maintained.
(3)$$\%\;Total\;viable\;RBC=\left[\frac{Number\;of\;viable\;RBCs}{Total\;number\;of\;RBCs\;counted}\right]\times 100$$

### 3.6. In Vitro Cytotoxic Activity in HepG2 Cells 

To determine the cytotoxic activity of nanoparticles, HepG2 cells were seeded in 48-well plates with a density of 1 × 10^4^ cells/well. Cells were maintained in DMEM supplemented with 10% FBS at 37 °C in a humidified and 5% CO_2_ incubator. After 48 h, the medium was replaced with fresh serum-free DMEM containing nanoparticle loaded with of 0–10 μg/mL of Dox or free Dox 0–10 μg/mL for an additional another 24 h. As a control was evaluated nanoparticles without Dox at different concentrations (0–200 μg/mL). Then, MTT (5 mg/mL) was added and the cells were incubated for 4 h. The formazan crystals formed were re-suspended with isopropanol and the absorbances were read at 570 nm [47]. The cytotoxic activity was reported as IC50 values.

### 3.7. Specific Recognition Assays

#### 3.7.1. Enzyme-Linked Lectin Recognition Assays (ELLA)

The interaction of *Ricinus communis* agglutinin I (RCA I) with lactosylated nanoparticles was evaluated by ELLA [24]. Briefly, 50 μg/100 μL lactosylated nanoparticles and negative controls (non-lactosylated nanoparticles) were immobilized in a 96-well plate and incubated for 24 h. After extensive washes with PBS, wells were blocked for 3 h with PBS containing, 0.05% Tween 20, pH 7.5 (PBS-T), and 1.5% BSA, to prevent non-specific interactions. Then, 100 μL of biotin-labeled RCA I (2.5 μg/mL) was added and incubated for 3 h. After washed with PBS-T, samples were incubated with 100 μL of streptavidin-peroxidase (1:2000) in PBS for 40 min. The color reaction was developed at room temperature by adding of peroxidase substrate, 0.075% 3,3′-diaminobenzidine-4HCl (DAB) and absorbance at 450 nm were read at 10 min.

#### 3.7.2. Evaluation of Specific Recognition by HepG2 Cells

Specific recognition of lactosylated nanoparticles by ASGPR of HepG2 cells was evaluated by the measuring of the DOX excitation and emission (λexc 480 and λemi 560 nm, respectively) [24]. HeLa cells were used as a negative control. Cells were seeded in 48 well plates at a density of 10,000 cells/200 μL per well using DMEM containing 10% FBS. After incubation (5% CO_2_ at 37 °C for 24 h), cells were washed three times with physiological saline solution (200 μL) and subsequently incubated with PBS containing different types of nanoparticles (lactosylated and non-lactosylated) at a 10 μg/200 μL at 37 °C for 30 min. After, the medium was removed, and cells were washed three times with physiological saline solution (200 μL) to remove unbound nanoparticles before observation. For competition assays, cells were incubated simultaneously with nanoparticles (10 μg/100 μL) and free lactose (10 μg/100 μL); for uptake inhibition assays, cells were preincubated with lactose (20 μg/200 μL) and washed three times with physiological saline solution (200 μL) before the nanoparticles (10 μg/200 μL) were added. The images were obtained by confocal microscopy (Nikon TiEclipse C2+, Japan) with 488-nm lasers at 20× magnification. The image dimensions were 1024 × 1024 pixels.

### 3.8. Statistical Analysis

Data are presented as mean ± standard deviation. Significant differences between treatments and control were determined by one-way analysis of variance (ANOVA) test and then by multiple comparisons between treatments using the Tukey’s test. *p* ≤ 0.05 were considered statistically significant.

## 4. Conclusions 

In this work, we describe the synthesis and characterization of nanoparticles of LC tBSA/BSALac-Dox NPs and HC tBSA/BSALac-Dox NPs. The nanoparticles loaded with Dox showed to be very stable, in addition to high encapsulation percent. Moreover, it was shown that the nanoparticle formulation is biocompatible with RBCs and liver cells. In the cytotoxicity assays, the success of loading the Dox in nanoparticles was shown, as opposed to being in free form. Finally, biorecognition, competition and inhibition assays demonstrated that the presence of galactose residues in nanoparticles confers them the capability to be efficiently recognized by HepG2 cells, specifically by ASGPR. Therefore, LC tBSA/BSALac-Dox NPs and HC tBSA/BSALac-Dox NPs may be used as a potential DDS for the liver’s targeted delivery.

## Figures and Tables

**Figure 1 molecules-25-05432-f001:**
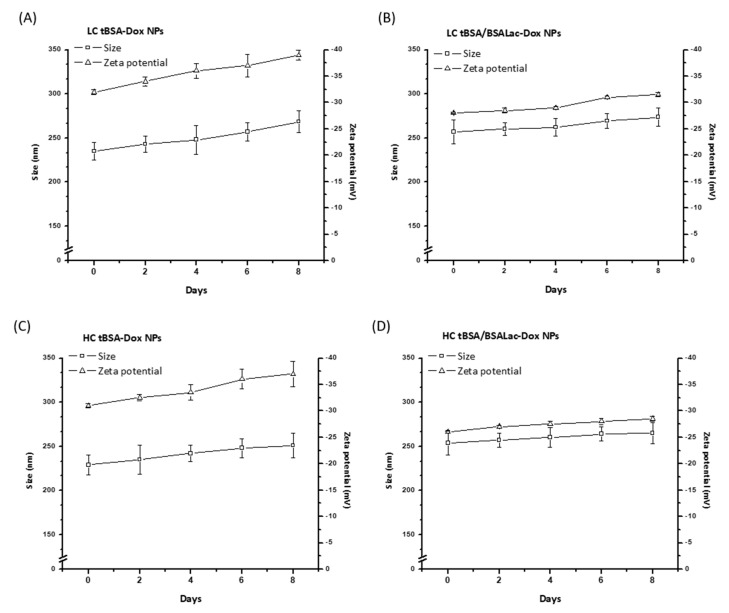
Analysis of nanoparticle stability in PBS pH 7.2 for 8 days by size and ζ−potential. (**A**) LC tBSA-Dox NPs; (**B**) LC tBSA/BSALac-Dox NPs; (**C**) HC tBSA-Dox NPs and (**D**) HC tBSA/BSALac-Dox NPs. Values are average and standard deviation for triplicate.

**Figure 2 molecules-25-05432-f002:**
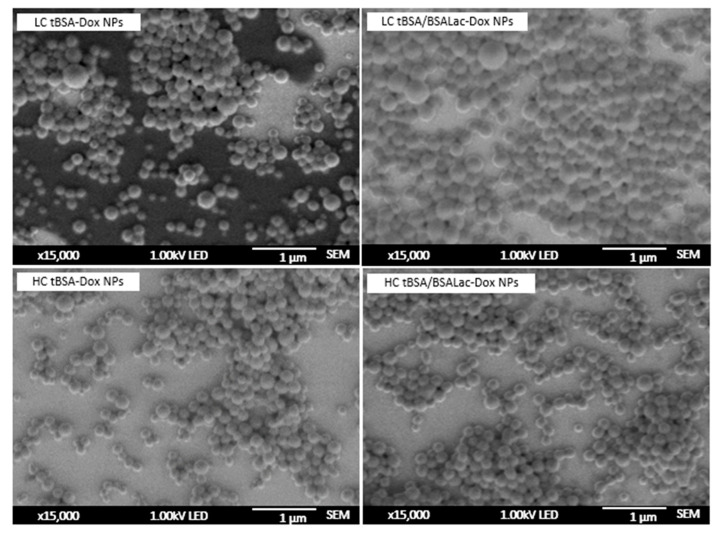
Analysis of nanoparticle morphology by scanning electron microscopy (SEM).

**Figure 3 molecules-25-05432-f003:**
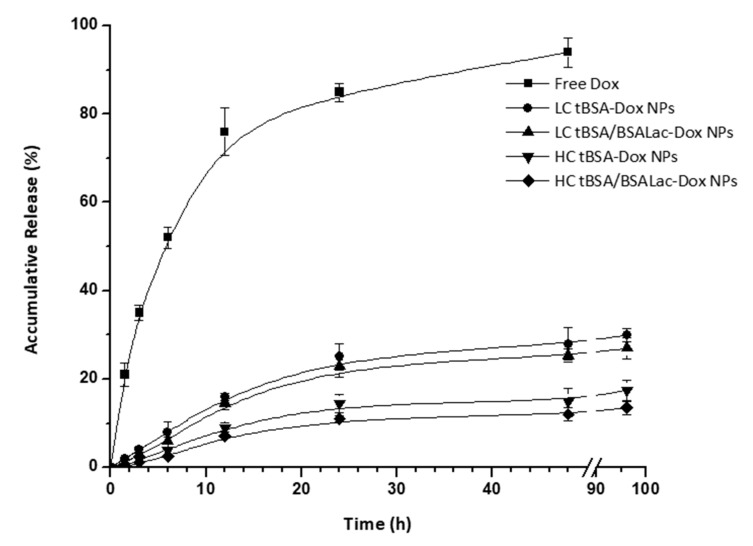
Accumulative release of Dox from Dox loaded nanoparticles in pH 7.2 PBS buffer. The free Dox was used as a control. Values are average and standard deviation for triplicate.

**Figure 4 molecules-25-05432-f004:**
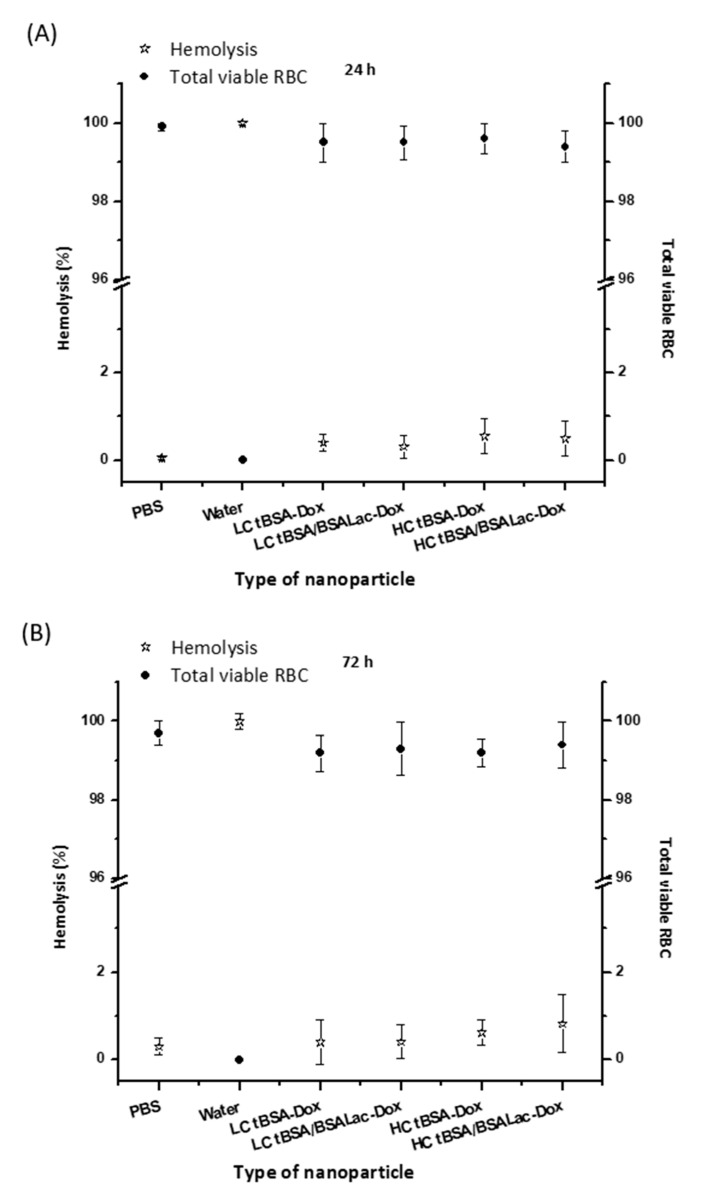
Percentage of hemolysis and RBCs viability induced by nanoparticles, (**A**) over 24 h and (**B**) 72 h incubation. The nanoparticle concentrations were 500 μg/mL. PBS and water were used as a control. Values are average and standard deviation for triplicate.

**Figure 5 molecules-25-05432-f005:**
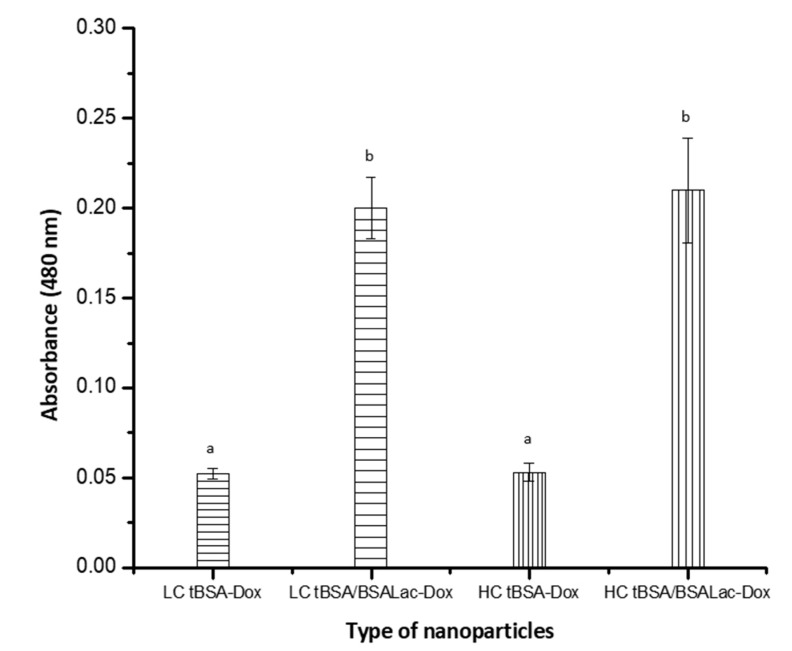
ELLA for the biorecognition of different types of nanoparticles by *Ricinus communis* lectin. Values are average and standard deviation for triplicate. The analysis was by one-way ANOVA followed by Tukey’s test. Different letters (a and b) indicate statistical differences (*p* ≤ 0.05).

**Figure 6 molecules-25-05432-f006:**
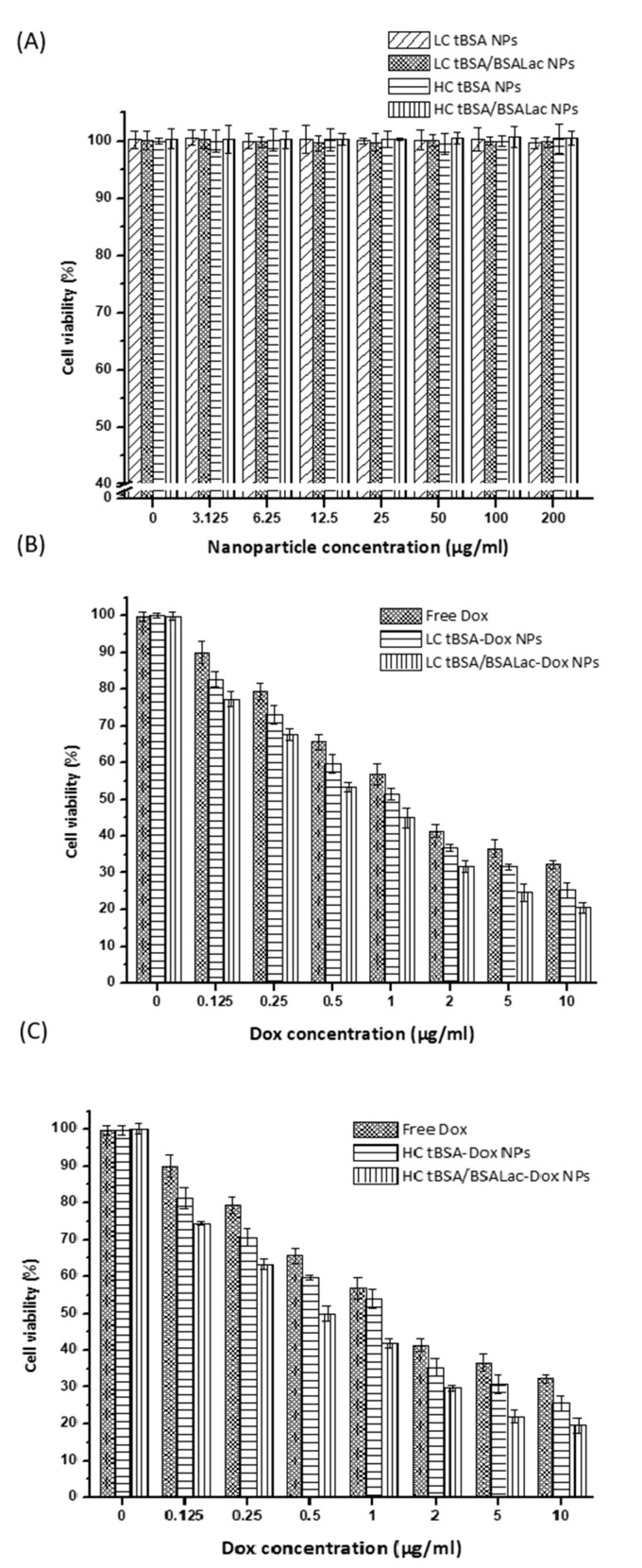
In vitro antitumoral activity in HepG2 cells of Dox from Dox loaded nanoparticles. (**A**) Nanoparticles without Dox, Dox loaded nanoparticles with (**B**) LC and (**C**) HC interacting with HepG2 cells. In B y C, free Dox was used as a positive control. Values are average and standard deviation for triplicate.

**Figure 7 molecules-25-05432-f007:**
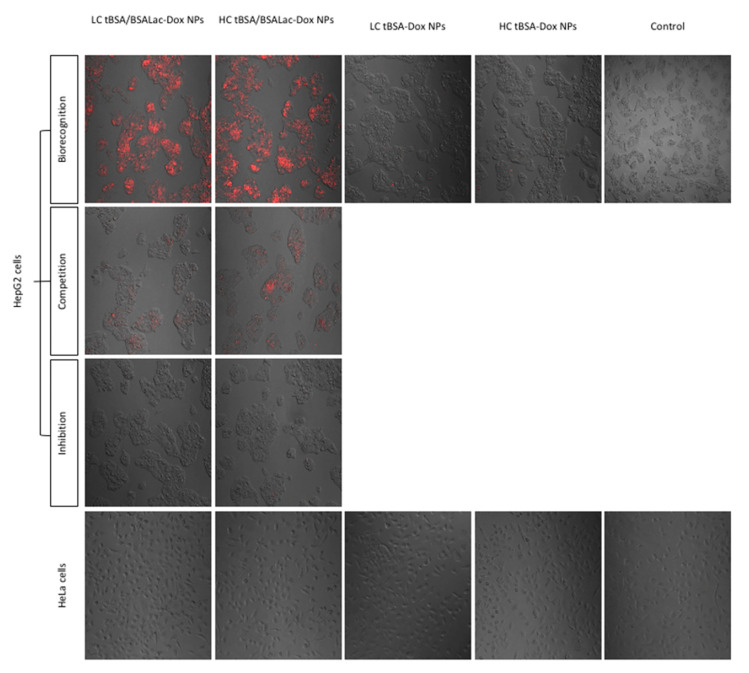
Confocal fluorescence images of the cellular recognition of Dox (red) loaded nanoparticles and their specific biorecognition by HepG2 cells. To confirm specific biorecognition competitive and inhibition assay were performed. In the competition assay, HepG2 cells were simultaneously incubated with nanoparticles and free lactose. In inhibition assay, HepG2 cells were first incubated with lactose and later nanoparticles were added. HeLa cells were used as a negative control.

**Figure 8 molecules-25-05432-f008:**
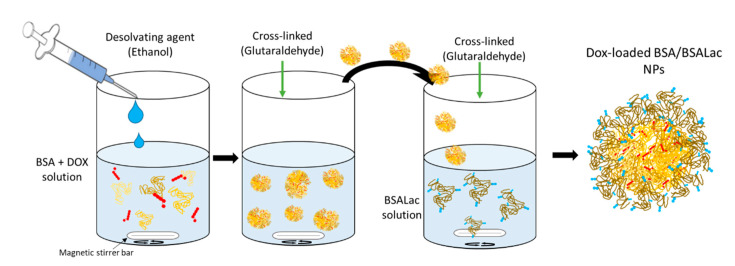
Schematic illustration of the process to obtain Dox-loaded albumin-albumin/lactosylated nanoparticles.

**Table 1 molecules-25-05432-t001:** Size, PDI, zeta potential and encapsulation efficiency of the nanoparticles.

	Nanoparticles			
	LC tBSA-Dox	LC tBSA/BSALac-Dox	HC tBSA-Dox	HC tBSA/BSALac-Dox
Size (nm)	235 ± 10	257 ± 14	229 ± 11	254 ± 14
PDI	0.17 ± 0.0	0.14 ± 0.1	0.18 ± 0.0	0.13 ± 0.1
Zeta potential (mV)	−32.0 ± 0.5	−28.0 ± 0.1	−31.0 ± 0.5	−26.0 ± 0.2
E.E. %	71.8 ± 1.3	73.4 ± 0.8	89 ± 2	91 ± 2

PDI = Polydispersity index and E.E. = Encapsulation efficiency. Values are average and standard deviation (±) for triplicate.

**Table 2 molecules-25-05432-t002:** In vitro antitumoral activity in HepG2 Cells of different nanoparticles with/without Dox.

HepG2 Cells			
Nanoparticles with Dox	IC_50_ (µg/mL)	Nanoparticles without Dox	IC_50_ (µg/mL)
LC tBSA-Dox	1.05 ± 0.13 ^a^	LC tBSA	ND
LC tBSA/BSALac-Dox	0.7 ± 0.09 ^a^	LC tBSA/BSALac	ND
HC tBSA-Dox	1.04 ± 0.15 ^a^	HC tBSA	ND
HC tBSA/BSALac-Dox	0.59 ± 0.07 ^a^	HC tBSA/BSALac	ND
Free Dox *	1.90 ± 0.42 ^b^		

IC_50_ values are average and standard deviation (±) for triplicate. Different letters (a and b) indicate statistical differences (*p* ≤ 0.05). ND: IC_50_ not determined at less than 200 μg/mL. * free Dox in PBS solution was used as a control.

**Table 3 molecules-25-05432-t003:** Synthesis of LC tBSA-Dox, LC tBSA/BSALac-Dox, HC tBSA-Dox and HC tBSA/BSALac-Dox NPs.

Type of Nanoparticles	Amount of Glutaraldehyde	BSA-Lac Shell
LC tBSA-Dox (Control)	5 μL	No
LC tBSA/BSALac-Dox	5 μL	Yes
HC tBSA-Dox (Control)	10 μL	No
HC tBSA/BSALac-Dox	10 μL	Yes

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
