# Peer review of "Albumin-Albumin/Lactosylated Core-Shell Nanoparticles: Therapy to Treat Hepatocellular Carcinoma for Controlled Delivery of Doxorubicin"

_molecules, 2020, doi:10.3390/molecules25225432_

Round 1

Reviewer 1 Report

In the paper entitled “Albumin-albumin/lactosylated core-shell nanoparticles: Therapy to treat hepatocellular carcinoma for controlled delivery of doxorubicin”, the authors described the synthesis process and the characterization of albumin/lactosylated albumin core-shell NPs (tBSA/BSALac NPs), loaded with doxorubicin (Dox). Hemocompatibility, cytotoxicity, and specific cell uptake were evaluated. The drug delivery system proposed in this paper is an evolution of lactosylated albumin NPs (BSALAc NPs) described by authors in ref 20.

In my opinion, the paper could be of interest to the readers of Molecules but major revisions are required before it could be reconsidered for publication.

A scheme showing the synthesis steps could be useful for a better understanding of the process. A scheme of the final drug delivery system could be also useful: reading the text, it’s not immediately clear the core-shell system, so as well the surface functional groups influencing the zeta potential; the residues used for the specific recognition could be also highlighted.

All data reported in Table 1 (and in the text of the manuscript) must be corrected reporting the errors with only one significant digit. The EEs are very high, authors should better comment on these data comparing them with similar results reported in the literature.

Figure 1 could be made more readable by increasing the label size; in the text and in the caption, it should be reported that the DLS measurements were made in PBS pH 7.2

The release curves seem to change their slops after about 12 hours. This aspect should be commented and release rates should be calculated and reported in the paper.

Please, increase the label size also in Figures 4 and 6. The absorbance values reported in section 2.4.1 should have the errors with only one significative digit.

The role of the crosslinker amount in DDS behavior should be better commented on; it seems to influence the EE and the release rates, but not the IC50 values (0.70 ± 0.09 and 0.59 ± 0.07) that are comparable within the errors.   

Please, substitute “Dox concentration” with “NPs concentration” in Figure 6 A.

Figure 7 reports confocal images of cells after 30 min of incubation with NPs. Usually, after 30 min, NPs greater than 200 nm interact with the cell membrane but they are not yet internalized. The authors should demonstrate the cell uptake reporting confocal images where cell nuclei and membranes are colored.

Reviewer 2 Report

In this manuscript, authors present the production and characterization of Dox-loaded albumin-albumin/lactosylated nanoparticles  (tBSA/BSALac  NPs), the cytotoxic activity of these nanoparticles on human liver cancer cells (HepG2 cells) which they target specifically. This is an interesting and well done study.

Here are my comments:

Abstract : indicate to what are crosslinked the nanoparticles, it is not obvious for me.

Introduction:

  • “It is specifically effective …” References 1 and 5 are 2003 and 1998…. List only those cancers for which dox is used today.
  • Correct “Passive targeting is mainly possible largely possible by the enhanced vascular permeability and retention (EPR).”
  • “Previously, we reported on the synthesis and characterization of lactosylated (Lac) albumin (BSA) nanoparticles (BSALac NPs), in addition to their specific recognition by the asialoglycoprotein receptor (ASGPR), which is practically exclusively and the most abundant receptor in liver cells [20].” If ASGPR is present on healthy liver cells, BSALac NPS will deliver dox to healthy liver cells and alter them. Can you comment on that?

Results:

  • Figure 4: Inversion between point and star in the "water" condition
  • Figure 7: I’m surprised that LC tBSA-Dox NPs and HC tBSA-Dox NPs showed very low biorecognition by HepG2 cells while they show a cytotoxicity for these cells (Table 2 and Fig. 6). Can you comment?

Reviewer 3 Report

The authors report DDS targeting hepatocellular carcinoma (HCC), specifically
to the asialoglycoprotein receptor (ASGPR). Dox-loaded albumin albumin/lactosylated (core-shell) nanoparticles (tBSA/BSALac NPs) with low (LC) and high (HC) crosslinking. Nanoparticles presented spherical shapes with a size distribution of 257 ± 13.8 nm and 254 ± 14.2 nm, as well as an estimated surface charge of −28 ± 0.11 mV and −26 ± 0.23 mV, respectively. The
encapsulation efficiency of Dox for the two types of nanoparticles was higher than 80%. The in vitro drug release results showed a sustained and controlled release profile. 

I would ask the authors to make the following mandatory changes to make the paper strong:

  1. The introduction can be improved for a broader audience. The authors should reference some other nanoparticle papers in killing cancer cells: for example Shao et al., 2007/7/6 Journal Nanotechnology Volume 18 Issue 31 Pages 315101; Panchapakesan et al., Nanomedicine 6 (10), 1787-1811; Panchapakesan et al., Nanobiotechnology 1 (2), 133-139; Loeian et al., Lab on a Chip 19 (11), 1899-1915. These papers should be referenced to make it more interesting in different types of drug delivery for killing cancer cells.
  2. Does the cytotoxicity of the particles as a function of particle size? More elaboration is needed here.
  3. What is the mechanism of drug delivery? Please elaborate on this?
  4. A discussion would be good on the results?
  5. What is the efficacy of cell uptake? I do not see a Cell viability assay?

I think this is a good paper. These changes are mandatory and will strengthen the paper and make it more interesting.

Round 2

Reviewer 1 Report

The paper was improved but some issues should still be addressed to make the work stronger.

-please, substitute in the abstract:

257 ±14

254 ± 14

-28.0 ± 0.1

-26.0 ± 0.2

-please, substitute in section 2.1.1

71.8 ± 1.3

73.4 ± 0.8

89 ± 2

91 ± 2

-please, correct the errors of data reported in Table 1.

- I cannot see the graphical abstract, can the author upload it?

A discussion on the surface functional groups influencing the zeta potential values, and on the residues used for the specific recognition is still missing.

Reviewer 3 Report

The authors should consider revising the paper on the lines indicated. None of the questions has been answered. 
